# RISE AND DISE: TWO FRAMEWORKS FOR LEARNING FROM TIME SERIES WITH MISSING DATA

## ABSTRACT

Time series with missing data constitute an important setting for machine learning. The most successful prior approaches for modeling such time series are based on recurrent neural networks that learn to impute unobserved values and then treat the imputed values as observed. We start by introducing *Recursive Input and State Estimation* (RISE), a general framework that encompasses such prior approaches as specific instances. Since RISE instances tend to suffer from poor long-term performance as errors are amplified in feedback loops, we propose *Direct Input and State Estimation* (DISE), a novel framework in which input and state representations are learned from observed data only. The key to DISE is to include time information in representation learning, which enables the direct modeling of arbitrary future time steps by effectively skipping over missing values, rather than imputing them, thus overcoming the error amplification encountered by RISE methods. We benchmark instances of both frameworks on two forecasting tasks, observing that DISE achieves state-of-the-art performance on both.

## 1 INTRODUCTION

Many machine learning settings involve data consisting of time series of observations. Due to various reasons, observations may be missing from such time series. For instance, it may be impossible to observe the data during a given time window, the data-recording system may fail, or measurements may be recognized as noisy and immediately discarded at the source. Historically, the prevalent approach to handling missing data has been to apply a preprocessing step to replace the missing observations with substituted values and then treat the time series as though it were complete (Schafer & Graham, 2002). Multiple recent works (Lipton et al., 2016; Yoon et al., 2017; 2018; Che et al., 2018; Cao et al., 2018), however, circumvent this two-step approach and integrate both value imputation and the downstream task in one single model. At their core, these approaches employ recurrent neural networks (RNN) (Hochreiter & Schmidhuber, 1997) whose input and hidden states are modified to account for the missing observations.

In this paper, we note fundamental commonalities between these prior approaches and define a general framework, *Recursive Input and State Estimation* (RISE), that encompasses those approaches as specific instances. Instances of RISE operate recursively on all intermediate time steps between two observed values, by imputing the first unobserved value in a row based on the preceding observed values and then working with the imputed value as though it were observed. Recursive approaches suffer from poor long-term performance (Lamb et al., 2016; Fox et al., 2018) as errors are amplified in feedback loops. To overcome this problem, we propose a novel framework, *Direct Input and State Estimation* (DISE), also built on recurrent neural networks. In contrast to RISE, DISE learns representations for inputs and hidden states that are enriched with time information and thus allow the framework to model arbitrary future time steps by effectively skipping over missing values, rather than imputing them. Moreover, unlike previous work (Yoon et al., 2017; Baytas et al., 2017; Binkowski et al., 2018; Che et al., 2018), DISE integrates not only relative time information (the time between observations), but also absolute time information, which helps the model learn non-stationary properties of the signal. Overall, we make the following contributions:

(i) We introduce RISE (Section 3), a general framework for learning from univariate time series with missing data, which captures the fundamental commonalities between the most relevant recursive approaches from the literature.

(ii) We propose DISE (Section 4), a non-recursive framework that learns time-enriched input and state representations and is thus able to avoid imputation-induced drift (a main shortcoming of RISE methods) by skipping to future hidden states.

(iii) We benchmark instances of RISE and DISE in two forecasting problems (Section 5). The best-performing instance of DISE operates on digits as atomic units to derive latent representations for numerical data. The expressiveness of this encoding function allows us to achieve state-of-the-art performance in both problems.

## 2 NOTATION

We denote a univariate time series $\boldsymbol{x} = [x_1, x_2, \ldots, x_T]$ as a sequence of $T$ observations. Let $\boldsymbol{s}$ be the time vector $[s_1, s_2, \ldots, s_T]$, each value corresponding to the time—measured as the running time elapsed since the beginning of the time series—when the observation was taken. In practice, some observations may be missing for multiple reasons, for which a $T$-dimensional masking vector $\boldsymbol{m}$ is defined: $m_i = 1$ if $x_i$ is observed and $m_i = 0$ otherwise.

We define an alternative notation that will be used in the definition of our framework. We denote $\boldsymbol{o} = [o_1, o_2, \ldots, o_N]$ as the sequence of $N$ observations with non-missing values. The associated timestamp vector is denoted by $\boldsymbol{t} = [t_1, t_2, \ldots, t_N]$, where $t_i$ represents the time—also represented as the running time since the beginning of the time series—of the observed value $o_i$. Moreover, we also define $\boldsymbol{t}^a$, which translates the running time values of $\boldsymbol{t}$ to a certain date or time format. In absence of missing data, $\boldsymbol{o} = \boldsymbol{x}$ and $N = T$.

A time series with missing data can be described with either notation (see Table 5 in Appendix A).

## 3 RECURSIVE INPUT AND STATE ESTIMATION

Instances of the RISE framework are given by recurrent neural architectures to learn a model for predicting $p(x_{i+n}|\boldsymbol{x}_{1:i})$ by recursively predicting intermediate conditional terms. While one can find a numbers of variants of recurrent neural architectures in the literature, all variants define a cell whose (hidden) state $\boldsymbol{h}_i \in \mathbb{R}^{d_h}$—$d_h$ is the number of units—updates based on the previous state $\boldsymbol{h}_{i-1}$ and current input $x_i \in \mathbb{R}$. In time series with missing data, instances of the RISE framework replace the standard input with a transformed input $\hat{x}$. Similarly, at the input of the cell the incoming state $\boldsymbol{h}_{i-1}$ is substituted with a transformed hidden state $\hat{\boldsymbol{h}}_{i-1}$ to account for the last time a value was observed in the time series. The hidden state $\boldsymbol{h}_i$ is updated based on these transformed signals and the equations of the chosen recurrent architecture. The transformed input $\hat{x}$ depends on the conditionally imputed input $x^c$, defined as follows:

$$x_i^c = x_i m_i + (1 - m_i)\tilde{x}_i, \tag{1}$$

where $\tilde{x}$ is the imputed input, whose computation is instance-specific.

It is common to some of these works to define a time gap vector $\boldsymbol{\delta}^B$, defined as the time gap from the last observed value to the current timestamp. More formally:

$$\delta_i^B = \begin{cases} s_i - s_{i-1} + \delta_{i-1}^B, & i > 1, m_{i-1} = 0 \\ s_i - s_{i-1}, & i > 1, m_{i-1} = 1 \\ 0, & i = 1 \end{cases} \tag{2}$$

The superscript $B$ refers to the backward computation of the time gap vector: the time gap at a certain moment is computed with respect to the previous timestamp. They are used in the computation of the so-called discount factors, denoted as $\gamma^x \in \mathbb{R}$ and $\gamma^h \in \mathbb{R}^{d_h}$, which in turn are used in the computation of the transformed state and the transformed hidden state, respectively. The discount factors are defined as follows:

$$\begin{aligned} \gamma_i^x &= \gamma^x(\delta_i^B) = \exp(-\max(0, w_\gamma^x \delta_i^B + b_\gamma^x)), \\ \gamma_i^h &= \gamma^h(\delta_i^B) = \exp(-\max(0, \boldsymbol{w}_\gamma^h \delta_i^B + \boldsymbol{b}_\gamma^h)), \end{aligned} \tag{3}$$

where $w_\gamma^x$, $b_\gamma^x$, $b_\gamma^h \in \mathbb{R}$ and $\boldsymbol{w}_\gamma^h \in \mathbb{R}^{d_h}$, are parameters trained jointly with all other parameters of the model. The motivation of the discount factor comes from the following intuition: the influence of the past history in the current moment fades away over time. These discount factors allow these models to also learn from time series where consecutive timestamps are not always equally spaced.

While in most of the RISE instances authors address related downstream time series classification or regression problems in multivariate time series, in this paper we are interested in autoregression problems in univariate time series, which allow to assess more directly the ability of these methods to learn with missing data. Therefore, a function $f_\theta$, parameterized with weights $\boldsymbol{\theta}$, is applied to the hidden state $\boldsymbol{h}_i$ to predict the next observation, more formally: $\hat{y}_{i+1} = f_\theta(\boldsymbol{h}_i)$. The loss at the $i$-th observation is defined as $l_i = m_{i+1}\mathscr{L}(y_{i+1}, \hat{y}_{i+1})$.

Typically, the function $f_\theta$ is a regression function, $\mathscr{L}$ is the mean squared error loss, and $y_i$ amounts to $x_i$—*i.e.* the $i$-th value of the time series. Alternatively, it can be cast as a multi-class classification problem (van den Oord et al., 2016b;a; Fox et al., 2018), wherein the function $f_\theta$ outputs a probability mass function, $\mathscr{L}$ is the cross-entropy loss, and $y_i$ is the one-hot encoding of $x_i$.

A non-exhaustive list of recent instances of the RISE framework can be found in Table 1.

| Instance of RISE | Input and State Transformation (Univariate Time Series) | | |
|---|---|---|---|
| Simple Recursion | $\tilde{x}_i = \boldsymbol{w}_x \cdot \boldsymbol{h}_{i-1} + b_x$ | $\hat{x}_i = x_i^c$ | $\hat{\boldsymbol{h}}_{i-1} = \boldsymbol{h}_{i-1}$ |
| Zeros&Indicators | $\tilde{x}_i = 0$ | $\hat{\boldsymbol{x}}_i = [x_i^c; \overline{m}_i]$ | $\hat{\boldsymbol{h}}_{i-1} = \boldsymbol{h}_{i-1}$ |
| Forward-Filling&Indicators | $\tilde{x}_i = x_{i'}$ | $\hat{\boldsymbol{x}}_i = [x_i^c; \overline{m}_i]$ | $\hat{\boldsymbol{h}}_{i-1} = \boldsymbol{h}_{i-1}$ |
| GRU-D | $\tilde{x}_i = \gamma_i^x x_{i'} + (1-\gamma_i^x)x_{av_{i'}}$ | $\hat{\boldsymbol{x}}_i = [x_i^c; m_i]$ | $\hat{\boldsymbol{h}}_{i-1} = \gamma_i^h \odot \boldsymbol{h}_{i-1}$ |
| RITS-I | $\tilde{x}_i = \boldsymbol{w}_x \cdot \boldsymbol{h}_{i-1} + b_x$ | $\hat{\boldsymbol{x}}_i = [x_i^c; m_i]$ | $\hat{\boldsymbol{h}}_{i-1} = \gamma_i^h \odot \boldsymbol{h}_{i-1}$ |
| (Luo et al., 2018) | $\tilde{x}_i = 0$ | $\hat{x}_i = x_i^c$ | $\hat{\boldsymbol{h}}_{i-1} = \gamma_i^h \odot \boldsymbol{h}_{i-1}$ |
| (Kim & Chi, 2018) | $\tilde{x}_i = \mathbf{1}_{\gamma_{i''}>\tau} x_{i''} + (1-\mathbf{1}_{\gamma_{i''}>\tau})x_{av}$ | $\hat{x}_i = x_i^c$ | $\hat{\boldsymbol{h}}_{i-1} = \boldsymbol{h}_{i-1}$ |
| (Yoon et al., 2017)* | $\tilde{x}_i = 0$ | $\hat{\boldsymbol{x}}_i = [x_i; \delta_i^B]$ | $\hat{\boldsymbol{h}}_{i-1} = \boldsymbol{h}_{i-1}$ |
| BRITS-I* | $\tilde{x}_i = \boldsymbol{w}_x \cdot \boldsymbol{h}_{i-1} + b_x$ | $\hat{\boldsymbol{x}}_i = [x_i^c; m_i]$ | $\hat{\boldsymbol{h}}_{i-1} = \gamma_i^h \odot \boldsymbol{h}_{i-1}$ |

Table 1: (Upper) $\cdot$ is the dot product. $[a;b]$ is the concatenation of terms $a$ and $b$. $\odot$ is the element-wise multiplication. $\overline{m}$ is the complement of the binary value $m$. $x_{av_{i'}}$ is the average of the non-missing values prior to $s_i$ of vector $\boldsymbol{x}$. $x_{i'}$ is the value of the last time ($i' < i$) the signal was observed. (Lower) Instances to the data imputation problem are indicated with a grey background. Luo et al. (2018) is framed in a generative adversarial network wherein the full time series is observed before imputing the missing values. $\gamma_{i''}$ is a discount factor computed with respect to $x_{i''}$, which is the nearest observed value in either time direction, and $\tau$ is a threshold value. An asterisk * indicates that the described equations are computed in a bidirectional manner. Except in cases where it is a constant, $\tilde{x}$ is always treated as a variable.

**Simple Recursion (Fox et al., 2018)**   This is the simplest instance of the RISE framework. At each timestamp the model is fed with either the input value if there exists, or the imputed value, which is computed in the previous timestamp, if the input value is missing.

**Zeros/Forward-Filling&Indicators (Lipton et al., 2016)**   Lipton et al. (2016) considers two popular strategies to fill missing values for sequential data: forward- and zeros-filling. The former replaces the missing values with the last non-missing value observed in the sequence. The latter simply fills the missing values with zeros. The input is augmented with a binary indicator, which is set to one if the input value is missing, and set to zero otherwise. Thus, through their hidden state computations, the recursive neural network can use indicators of missing data to learn arbitrary functions of the past observations and missingness patterns. Zero-filling with binary indicators was found to be the best overall model by all considered metrics in their downstream task.

**GRU-D (Che et al., 2018)**   This approach incorporates the previously described discount factors to account for missingness when computing the transformed input value and the hidden state. The decay mechanism applied to the input learns to fade away the impact of the last observation over time and progressively replace it with the empirical mean of the input signal. Similarly when applied to the hidden state, their features are decayed over time.

**RITS-I (Cao et al., 2018)** Cao et al. (2018) proposed BRITS, a family of models designed for data imputation in time series. Within these models, RITS-I is the only one that is suitable for the task at hand. It combines aspects of all three previous works. Similar to the simple recursion approach, the imputed value for the current timestamp is the predicted value computed in the previous timestamp. As in Che et al. (2018), it is equipped with a decay mechanism over the hidden state, and the input is augmented with a binary indicator.

**Other instances** Kim & Chi (2018) is another instance of the RISE framework closely related to GRU-D. Recent works (Yoon et al., 2017; Kim et al., 2017; Luo et al., 2018) on data imputation for time series could also be framed in the RISE framework under certain (small) modifications. Different to forecasting, most of data imputation methods replace missing values with substituted values based on previous and subsequent information.

## 4 DIRECT INPUT AND STATE ESTIMATION

Main (and only) differences across RISE instances lie on the equations that alter the standard input and hidden state of the recurrent neural architecture, but everything else remain unchanged. One may define new equations based on expert knowledge or intuition to guide the transformation of input and hidden state, however the DISE framework skip timestamps where the data is missing and only learns from observed data. DISE resorts to latent representations enriched with absolute and relative time information to enable the recurrent architecture to update its hidden state so as to jump to a certain moment into the future. The learning of the underlying "discount" factor is up to these representations. Therefore, DISE does not require computing any intermediate term. This implies that the hidden state is only updated in timestamps where a value is observed. We resort to the alternative notation defined in Section 2 to define DISE.

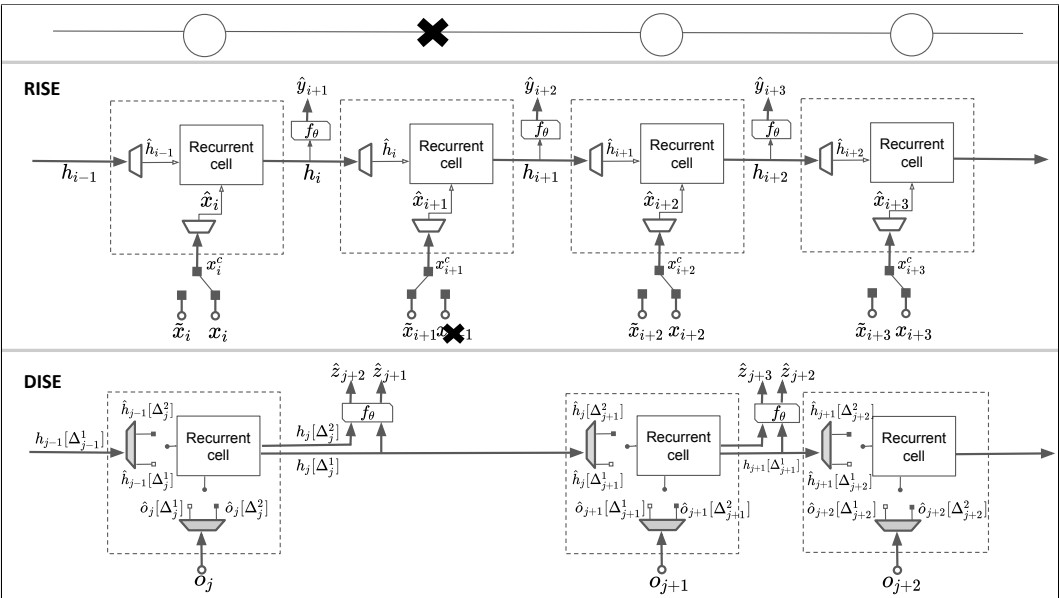

Figure 1: (Upper) A segment of a time series with missing data. Simplified layouts of (Middle) RISE and (Lower) DISE. Whereas RISE attends to timestamps whose data points are missing and updates its hidden state accordingly, DISE skips them and only updates its state with observed values.

The DISE framework also defines time gaps between pairs of non-missing observations as follows:

$$\delta_{i;n}^F = \begin{cases} t_{i+n} - t_i, & 1 \le i, i+n < N, \\ \text{undefined}, & \text{otherwise}, \end{cases} \tag{4}$$

which corresponds to the time gap between the current non-missing observation, happening at $t_i$, and the $n$-th non-missing observation after the current one, happening at $t_{i+n}$. In this case, the superscript $F$ refers to its forward computation.

Let $\Delta_i^n$ be a shortcut that refers to the time information $(\delta_{i;n}^F, t_{i+n}^a)$. DISE defines $\hat{o}_i[\Delta_i^n] \in \mathbb{R}^{d_o}$ and $\hat{h}_{i-1}[\Delta_i^n] \in \mathbb{R}^{d_h}$, computed respectively with functions $g_o$ and $g_h$, and that correspond to latent representations that are enriched with the absolute and relative time information in $\Delta_i^n$. They are the counterparts of the transformed input and state of the RISE framework. During training, at a certain time $t_i$ the framework computes:

- $\hat{o}_i[\Delta_i^n] = g_o(o_i, \delta_{i;n}^F, t_{i+n}^a)$ : The dedicated latent representation for the input $o_i$ to predict the value at $t_{i+n}$. The relative time distance between the two observations is $\delta_{i;n}^F$.

- $\hat{h}_{i-1}[\Delta_i^n] = g_h(h_{i-1}[\Delta_{i-1}^1], \delta_{i;n}^F, t_{i+n}^a)$ : The dedicated representation for the incoming hidden state $h_{i-1}[\Delta_{i-1}^1]$ to predict the value at $t_{i+n}$, separated $\delta_{i;n}^F$ from the the current observation.

We refer to $h_i[\Delta_i^n] \in \mathbb{R}^{d_h}$ as the $\Delta_i^n$-lagged hidden state, which will be used by the function $f_\theta$ to output $\hat{z}_{i+n}$—*i.e.* the prediction for $t_{i+n}$. It is computed based on $\hat{o}_i[\Delta_i^n]$ and $\hat{h}_{i-1}[\Delta_i^n]$, and the equations of the chosen recurrent architecture. One can train a model to be good at making predictions not only for the next observed value but also for observations beyond the next one or even for a time horizon. In the most general case, the loss at the $i$-th observation is defined as $l_i = \sum_{n=1}^{D} \mathscr{L}(z_{i+n}, \hat{z}_{i+n})$, where $D$ is the length—in number of observations—of the time horizon. As in the RISE framework, $\mathscr{L}$, $f_\theta$ and $z_i$ are chosen so as to cast the problem as either a regression or a multi-classification one. Figure 1 depicts the RISE and DISE frameworks. At inference time, we directly target a desired future timestamp $t_b^a$ and query the framework with $(t_b - t_a, t_b^a)$, being $t_a$ the timestamp of the last observed value, which computes the $(t_b - t_a, t_b^a)$-lagged hidden state to output a prediction.

One key component of the DISE framework is the learning of time-enriched input and hidden state representations. In this work we propose an instance whose functions $g_o$ and $g_h$ use the learned representations for relative and absolute time information to act as gating mechanisms over the representations of the input and hidden state. This allows to effectively allocate some dimensions of the representations to account for time dependencies. They are computed as follows

$$\hat{o}_i[\Delta_i^n] = g_o(o_i, \delta_{i;n}^F, t_{i+n}^a) = f_{\text{enc}}^\circ(o_i) \odot \underbrace{\sigma(W_\delta f_{\text{enc}}^{\delta^F}(\delta_{i;n}^F)) \odot \sigma(W_t f_{\text{enc}}^t(t_{i+n}^a))}_{\Phi^o(\Delta_i^n)}$$

$$\hat{h}_{i-1}[\Delta_i^n] = g_h(h_{i-1}[\Delta_{i-1}^1], \delta_{i;n}^F, t_{i+n}^a) = h_{i-1}[\Delta_{i-1}^1] \odot \underbrace{\sigma(V_\delta f_{\text{enc}}^{\delta^F}(\delta_{i;n}^F)) \odot \sigma(V_t f_{\text{enc}}^t(t_{i+n}^a))}_{\Phi^h(\Delta_i^n)} \tag{5}$$

where all $f^\Omega : \mathbb{R} \to \mathbb{R}^{d_\Omega}$ functions are encoders—we use the same $d_\Omega$ for all encoders—for numerical values. The $W, V$ matrices map all latent representations to the same dimension, either $d_o$ (which is the same as $d_\Omega$) or $d_h$. The choice of the encoders will be discussed in Section 4.1. The $\Phi^o(\Delta_i^n)$ and $\Phi^h(\Delta_i^n)$ terms have a role similar to the discount mechanisms of RISE, and are computed with both relative and absolute time information. However, these "discount" factors are to be learned, and not driven by a certain expression. Compared to the RISE framework, on one side DISE requires to compute additional latent representation for input, relative and absolute time information; but on the other side it does not require to compute intermediate states to make predictions for a future timestamp. This makes our framework also suitable for event series, where the time gap between two consecutive observations is arbitrary. In contrast, RISE instances will require events to be aligned at some chosen timestamps, which generates a large number of missing observations—thus, being inefficient computationally and increasing its complexity.

The representations for the input $\hat{o}_i[\Delta_i^n]$ and hidden state $\hat{h}_{i-1}[\Delta_i^n]$ are computed as the element-wise product of three representations. Alternatively, $g_o$ and $g_h$ could be chosen so as to represent $\hat{o}_i[\Delta_i^n]$ and $\hat{h}_{i-1}[\Delta_i^n]$ as the concatenation of their three corresponding representations. However, we experienced worse performance and longer training times with this instance, as the number of parameters of the recurrent cell also grow.

As discussed, instances of RISE and DISE are agnostic to the chosen recurrent neural architecture. In the experimental section we opt for a gated recurrent unit (GRU) (Cho et al., 2014), whose details for both frameworks can be found in Appendix B.1. An extension of the proposed DISE instance to increase its depth is explained in Appendix B.2. This extension reinforces time information into each of the higher layers. Note that $d_o$ might be different to $d_\Omega$ for higher layers in deeper architectures.

Most instances of RISE are not appropriate to DISE, as i) they cannot include (date-formatted) absolute time information, and ii) the transformed input and hidden state cannot not directly be computed for an arbitrary time into the future. However, it is possible to recycle ideas from RISE instances to only learn from observed data. Inspired by some of these works, we propose RISE2DISE, which applies corresponding discount mechanisms—see Equation (3)—to the input and hidden state to translate them to any moment into the future. Different to GRU-D, RISE2DISE only updates the state with observed values. More details to be found in Appendix B.3.

**Prediction over a Time Horizon**   Instances of the the RISE framework predict $x_{i+D}$ by recursively estimating intermediate values until reaching the desired point in time. Therefore, the side effect is that these methods also predict a time horizon—*i.e.* the values leading up to $x_{i+D}$. On the other side, multi-output forecasting methods estimate the whole time horizon in one step (Taieb et al., 2012). Previous work (Fox et al., 2018) has empirically shown that multi-output methods outperform recursive methods in the time horizon prediction task. This is due to the feedback loop of recursive methods, which re-use past predictions to predict future values, leading to lower quality predictions as the time horizon increases. However, it is not clear how multi-output approaches would deal with time series with missing values, as they assume the time series is complete. The DISE framework does not fit in either of these two paradigms.

## 4.1    ON THE CHOICE OF THE ENCODERS

Encoders suitable for DISE must be able to map any data that can be decomposed as a sequence of digits to a latent space. One may be tempted to transform these data to one-hot encoding representations, and then learn a latent representation via a linear layer. However, the potential vocabulary size of this approach is infinite. Therefore a data binning scheme should be first applied to the data, which is a challenge in itself.

**Feedforward-based encoders**   A popular option (Li et al., 2017; Pezeshkpour et al., 2018) for mapping single numerical values to a latent space is by applying a feedforward neural network over the (typically) log-normalized numerical value. Let $m$ and $\boldsymbol{m}$ be a numerical value and its corresponding latent representation, respectively, then this encoding function would amount to: $\boldsymbol{m} = f_{\mathtt{ffw}}(m) = \log(m)\boldsymbol{w}_{fw} + b_{fw}$, where the numerical value $m$ only scales the weight vector $\boldsymbol{w}_{fw}$. A sigmoid activation function is also added when encoding the input signal in Equation (5). Despite this non-linearity, the expressiveness of this encoding function is limited. It is trivial to show that the learned latent representations behave monotonically (with respect to $m$) because of the monotonicity of the sigmoid function and the linearity of the learned representations. To alleviate this limitation we also explore a multi-feedforward encoding function $f_{\mathtt{mffw}}$ with two layers and a sigmoid as an activation function.

**Digit-level encoder**   On the other hand, motivated by character-level architectures for language modeling, an alternative would consist of decomposing numerical values as a sequence of digits and then operate on digits as atomic units to derive latent representations. To our knowledge, using digits as atomic units (tokens) to learn representations for numerical values has been previously explored only in García-Durán et al. (2018) for a different problem—link prediction in temporal graphs. Each token of the numerical value is mapped to its corresponding embedding via a linear layer and the resulting sequence of embeddings is fed into a standard recurrent neural network architecture—a GRU in this work. Let $n_m$ be the number of tokens of the numerical value $m$, its latent representation corresponds to the last hidden state of the standard GRU, more formally: $\boldsymbol{m} = f_{\mathtt{gru}}(m) = \boldsymbol{h}_{n_m}$. In contrast to $f_{\mathtt{ffw}}$, the learned representations are not necessarily monotonic. The vocabulary consists of 11 tokens: for digits from 0 to 9, and token ".", which indicates, if there exists, the beginning of the decimal part of the numerical value. On some occasions, positive and negative signs may also be required in the vocabulary. Moreover, this encoding function allows absolute time information $t^a$ to be simply decomposed as a sequence of digits following a certain date format.

In any case, a dedicated encoding function is used for each of the input signal, relative and absolute time information (see superscripts in Equation (5)). We will refer to the instances whose encoding function is $f_{\mathtt{ffw}}$, $f_{\mathtt{mffw}}$ and $f_{\mathtt{gru}}$ as DISE-FFW, DISE-MFFW and DISE-GRU, respectively.

## 5 Experiments

We target two forecasting problems. For the time horizon prediction problem we consider time series without missing values, wherein the goal is to be good at predicting the values over a time horizon of length $D$. For the second one, referred to as next observed value prediction problem, we consider data sets with missing data. This means that the next observed value can happen anytime into the future, and the model makes a prediction for that specific time. They are illustrated in Figure 2.

Figure 2: (Upper) Time horizon and (Lower) next observed value prediction problem. Predictions (blue triangles) are made based on current (green squares) and past observed values (red circles). Black-filled shapes represent missing data.

### 5.1 Dataset Description

#### 5.1.1 Blood Glucose

This complete data (Fox et al., 2018) was collected over the course of three years and consists of a large number of continuous glucose readings from 40 patients with type 1 diabetes. Every three months, blood glucose of the patients was monitored over the course of several days with a sampling rate of 5 minutes. We do not use date-formatted time information $t^a$ in the experiments run in this data set, as many measurements lack a reference date. We run experiments in this data set for the time horizon prediction problem. We artificially remove consecutive blocks of data— see Appendix D.1—to the next observed value prediction problem, leading to the data sets Blood Glucose ($\beta = \{1,5\}$). The larger the value of $\beta$ is, the more data is missing. We use the same training, validation and test sets as in Fox et al. (2018).

#### 5.1.2 Air Pollution

This incomplete data (Yi et al., 2016) consists of measurements from monitoring stations in Beijing. They were hourly collected from 2014/05/10 to 2015/04/30. In this work we consider two particulate matter: $PM_{2.5}$ and $PM_{10}$. The first particle pollution time series was used in previous work (Yi et al., 2016; Cao et al., 2018) for the data imputation problem. We run experiments for the next observed value prediction problem in these data sets.

We have also created a modified version of the $PM_{2.5}$ time series, called $PM_{2.5-peak}$, wherein the measurements taken on the 2nd and 15th of each month are increased in $200 \, \mu g/m^3$—original values range from 1 to 1,000 $\mu g/m^3$. Therefore, the bias of the signal varies sharply in these days.

The first ten months of collected data are used for training, and the data collected in the 11th and 12th month are used for validation and test, respectively. Statistics of the data sets and additional details are shown in Table 5 in Appendix D.1.

| Time Horizon / Method | 5 Minutes | | 10 Minutes | | 15 Minutes | | 20 Minutes | | 25 Minutes | | 30 Minutes | |
|---|---|---|---|---|---|---|---|---|---|---|---|---|
| Simple Recursion (Fox et al., 2018) | **1.44** | 2.49 | 2.99 | 4.92 | 4.46 | 6.97 | 5.80 | 8.84 | 7.09 | 10.61 | 8.34 | 12.30 |
| DEEPMO (Fox et al., 2018) | 1.62 | 2.64 | 2.93 | 4.85 | 4.23 | 6.78 | 5.47 | 8.57 | 6.68 | 10.29 | 7.87 | 11.97 |
| SEQMO (Fox et al., 2018) | 1.50 | 2.52 | 2.78 | 4.72 | 4.08 | 6.66 | 5.28 | 8.45 | 6.50 | 10.18 | 7.63 | 11.83 |
| DISE-FFW [R] | 1.72 | 2.77 | 2.90 | 4.88 | 4.16 | 6.81 | 5.34 | 8.56 | 6.48 | 10.19 | 7.60 | 11.77 |
| DISE-MFFW [R] | 1.73 | 2.78 | 2.90 | 4.84 | 4.16 | 6.71 | 5.40 | 8.47 | 6.57 | 10.08 | 7.60 | **11.58** |
| DISE-GRU [R] | **1.44** | **2.48** | **2.72** | **4.68** | **4.00** | **6.57** | **5.15** | **8.31** | **6.34** | **10.01** | **7.51** | 11.62 |

Table 2: Blood Glucose: Time horizon prediction.

### 5.2 Setup and Baselines

As the blood glucose data set is our main benchmark data set, for all models we carefully follow the same evaluation protocol and setup as in Fox et al. (2018), where this data set was introduced. Therefore, instances of RISE are built by stacking two GRUs of 512 hidden units $d_h$ in a standard manner (Hermans & Schrauwen, 2013). Instances of the DISE framework are reinforced with time information at higher layers as explained in Appendix B.2. Following Fox et al. (2018), we use the multi-classification formulation of the problem described in Section 3 and the median and mean APE (absolute percentage error) as evaluation metrics. More training details are found in Appendix C. For the air pollution data sets the regression formulation replaces the multi-classification one, but every other aspect of the evaluation and setup remains the same. We think that this formulation is

more common in regression problems, and also helps to assess methods' performance under two different objectives.

For the time horizon problem, we evaluate our instance against the two multi-output approaches[1] proposed in Fox et al. (2018). We also compare it to the recursive approach previously referred to as simple recursion. The performance of these methods is copied from Fox et al. (2018). For the next observed value prediction problem we implemented most of the instances of the RISE framework (upper side in Table 1) and ran experiments under the above-described evaluation and setup. In neither of these two problems we compare to methods that are not based on recurrent neural architectures, as related work have shown they perform poorly compared to method based on recurrent architectures in related problems (Lipton et al., 2016; Fox et al., 2018; Cao et al., 2018).

For all instances of DISE we use a [*suffix*] that is a combination of the letters R (relative time information) and A (absolute time information) to indicate the use of that data in the learning of the $\Phi^o$ and $\Phi^h$ terms. For each data set, the $\delta^F$ values are normalized by the corresponding sampling period.

### 5.3 RESULTS

The format "Median APE | Mean APE" is used in all tables.

#### 5.3.1 BLOOD GLUCOSE

We show results for the time horizon problem in the blood glucose data set in Table 2. This was the problem this data set was originally intended for. As all baselines, our instances were trained for a length $D$ of the time horizon of 30 minutes—this

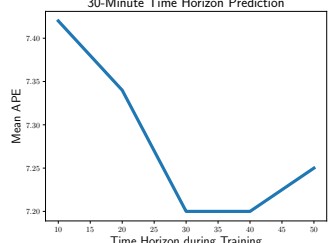

Figure 3: Mean APE *vs D*

amounts to 6 observations. DISE-GRU [R] shows the best performance for almost each of the time points within the 30-minute prediction window. The recursive approach matches the performance of DISE-GRU [R] at the first place of the time window. However, its performance degrades as the time horizon increases, being quickly surpassed by the multi-output approaches DEEPMO and SEQMO. The results also indicate the importance on the choice of the encoding function, as the two other instances of DISE are on par or even slightly worse than those of RISE.

One may wonder whether increasing the length $D$ of the time horizon during training may lead to better forecasts within the next 30-minute. Figure 3 shows the best performance is obtained when training and test objectives are pretty much aligned. As expected, performance degrades when the horizon of the training objective is smaller or larger (enough) than that of the evaluation.

Table 3 shows results for the next observed value prediction problem. Motivated by Figure 3, we align training and test objectives: models are trained to be good at predicting the next observed value. GRU-D is the best performing recursive method, and is competitive with DISE-FFW [R], but outper-

| Method \ Data Set | Blood Glucose ($\beta = 1$) | Blood Glucose ($\beta = 5$) |
|---|---|---|
| Simple Recursion | 2.43 \| 4.29 | 7.36 \| 12.43 |
| Zeros&Indicators | 2.43 \| 4.32 | 6.73 \| 11.64 |
| GRU-D | 2.25 \| 4.06 | 6.18 \| 11.53 |
| RITS-I | 2.32 \| 4.13 | 6.66 \| 12.23 |
| RISE2DISE | 2.48 \| 4.28 | 6.86 \| 11.98 |
| DISE-FFW [R] | 2.31 \| 4.09 | 6.02 \| 11.31 |
| DISE-MFFW [R] | 2.50 \| 4.29 | 6.31 \| 11.39 |
| DISE-GRU [R] | **2.06** \| **3.97** | **5.55** \| **11.29** |

Table 3: Blood Glucose: Next observed value prediction.

formed by DISE-GRU [R] in both mean and median APE. This indicates again that the choice of the encoding function is crucial to outperform the instances of RISE. RISE2DISE's performance indicates that exponential discount factors do not benefit from learning with only observed data. It is important to note that, contrary to language, similarities across numerical values based on its atomic units is arbitrary and based on the chosen notation scheme. We provide visualizations of the learned representations, and experiments with a hexadecimal notation scheme in Appendix D.2.

Once we have validated the choice of the encoding function, for the remaining experiments we only benchmark DISE-GRU against the baselines.

---

[1]The two other multi-output approaches proposed in that work are extensions of these ones, wherein the main difference lies on additional pre- and post-processing steps applied to the data.

### 5.3.2 AIR POLLUTION

The date-formatted timestamps $t^a$ are decomposed as the sequence of the digits that correspond to the month and

| Data Set
Method | Air Pollution - $PM_{10}$ | Air Pollution - $PM_{2.5}$ | Air Pollution - $PM_{2.5-peak}$ |
|---|---|---|---|
| Simple Recursion | 14.27 \| 32.52 | 12.29 \| 30.30 | 11.46 \| 33.16 |
| Zeros&Indicators | 14.57 \| 31.65 | 12.29 \| **28.22** | 11.46 \| 33.62 |
| GRU-D | 14.17 \| 28.85 | 12.37 \| 28.47 | 11.62 \| 32.49 |
| RITS-I | 14.42 \| 29.57 | 12.24 \| 28.68 | 11.19 \| 31.37 |
| DISE-GRU [R] | **13.81** \| 28.25 | **12.19** \| 28.94 | **11.12** \| 30.10 |
| DISE-GRU [AR] | **13.81** \| **27.60** | 12.44 \| 28.74 | 11.54 \| **27.76** |

Table 4: Air Pollution: Next observed value prediction.

the day of the month. Results are shown in Table 4. The time series corresponding to the particulate matters $PM_{10}$ and $PM_{2.5}$ do not largely benefit from including absolute time information. As before, GRU-D is the best performing recursive instance. For the particle $PM_{2.5}$, whose percentage of missing data is low, instances of RISE are competitive and even outperform instances of DISE in terms of mean APE. In this data set, majority (around 95%) of test data points are spaced one sampling period, so this may explain the very good performance of the RISE instances. One can assess the positive impact of the absolute time information in $PM_{2.5-peak}$, where DISE-GRU [AR] clearly outperforms all methods in mean APE. Figure 4 shows the improvement in performance occurs on the 2nd, 15th and their both subsequent days—highlighted with an orange background—, as it is in these days where the bias value of the time series sharply varies. We show in Appendix D.2 that DISE-GRU [AR] learns to allocate dedicated dimensions to model this characteristic of the signal. Even though recurrent neural networks are suitable to learn from non-stationary time series, this shows that including absolute time information helps to model certain properties of the signal—in this case, an abrupt change in the bias of the signal—that may be difficult to learn otherwise. To overcome this problem one might preprocess the time series to convert it into stationary, but this might be specially challenging under the existence of missing data in the time series.

## 6 DISCUSSION AND FUTURE WORK

We describe RISE, a recursion-based framework that encompasses most of the recent works in learning from time series with missing data. We propose a non-recursive counterpart, called DISE, based on latent representations that incorporates relative and absolute absolute time information. We show the benefits of an instance of this novel framework wherein latent representations for numerical data are derived from the digits they are made up of.

DISE relates to recent advances in the design of recurrent neural networks to learn long-term dependencies. Pachitariu & Sahani (2013); Chang et al. (2017) reinforce dependencies between non-consecutive states by adding connections between them. Different from these works, DISE does not require to manually create such connections. Rather, the model is able to jump to subsequent states because of its time-enriched representations. The continuous-time event prediction problem is a related topic. Event time series attend to sequences of actions that occur asynchronously. Examples are clinical visits of patients to the hospital or users purchasing history in an online marketplace. One major

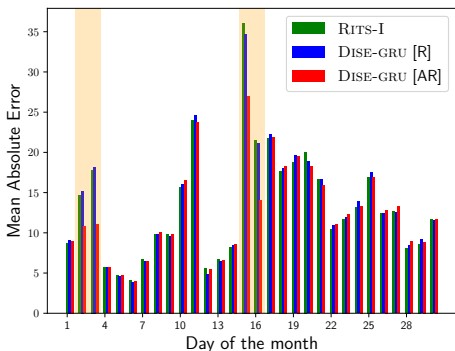

Figure 4: $PM_{2.5-peak}$: MAE vs day.

difference is that the input is always an event, which is typically a discrete token, and not a single continuous value. Solutions to this problem involve modifications of recurrent neural architecture (Neil et al., 2016; Zhu et al., 2017) or the use of some type of time representation that is typically concatenated to the event latent representation and used as the input to the recurrent architecture (Du et al., 2016; Mei & Eisner, 2017; Kazemi et al., 2019). Most related to our work is Li et al. (2017), wherein event embeddings are enriched with time information via the encoding function $f_{ffw}$. A second related topic is data imputation (Rubin, 1976). The major difference is that data imputation methods also use subsequent values to fill the missing values. The study of the DISE framework to these two related problems is left for future work. One might also want to evaluate DISE with recent recurrent architectures whose hidden state dynamics are specified by neural ordinary differential equations (Rubanova et al., 2019; De Brouwer et al., 2019). Most importantly, future work should make DISE also suitable for learning from multivariate time series.

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

## A  NOTATION: ONE EXAMPLE

An illustration of a time series under the two notations defined in Section 2 is depicted in Figure 5.

| Time Series $x$ | 47 | N.A. | N.A. | 40 | N.A. | 43 | 55 |
|---|---|---|---|---|---|---|---|
| Timestamps $s$ | 0 | 2 | 4 | 8 | 10 | 12 | 14 |
| Masking $m$ | 1 | 0 | 0 | 1 | 0 | 1 | 1 |

| Time Series $o$ | 47 | 40 | 43 | 55 |
|---|---|---|---|---|
| Timestamps $t$ | 0 | 8 | 12 | 14 |
| Date-Formatted Timestamps $t^a$ | 06:00 | 14:00 | 18:00 | 20:00 |

Figure 5: (Left) A univariate time series with missing values. (Right) Alternative notation for the same time series.

## B  RISE&DISE

### B.1  LEARNING WITH A GRU

As discussed, neither RISE not DISE are restricted to a particular type of recurrent network. As in prior work (Che et al., 2018), we opt for a gated recurrent unit (GRU).

Within the RISE framework, the equations that define the reset and update gates of the GRU unit to control the hidden state are as follows:

$$
\begin{aligned}
r_i &= \sigma(W_r \hat{x}_i + U_r \hat{h}_{i-1} + b_r) \\
z_i &= \sigma(W_z \hat{x}_i + U_z \hat{h}_{i-1} + b_z) \\
c_i &= \tanh(W \hat{x}_i + U(r_i \odot \hat{h}_{i-1}) + b) \\
h_i &= (1 - z_i) \odot \hat{h}_{i-1} + z_i \odot c_i,
\end{aligned}
\tag{6}
$$

where $\sigma$ is the sigmoid function, $\odot$ is the element-wise product, and all parameters are those of the standard setting where the GRU unit operates.

The GRU equations used in the DISE framework are as follows:

$$
\begin{aligned}
r_i[\Delta_i^n] &= \sigma(W_r \hat{o}_i[\Delta_i^n] + U_r \hat{h}_{i-1}[\Delta_i^n] + b_r) \\
z_i[\Delta_i^n] &= \sigma(W_z \hat{o}_i[\Delta_i^n] + U_z \hat{h}_{i-1}[\Delta_i^n] + b_z) \\
c_i[\Delta_i^n] &= \tanh(W \hat{o}_i[\Delta_i^n] + U(r_i[\Delta_i^n] \odot \hat{h}_{i-1}[\Delta_i^n]) + b) \\
h_i[\Delta_i^n] &= (1 - z_i[\Delta_i^n]) \odot \hat{h}_{i-1}[\Delta_i^n] + z_i[\Delta_i^n] \odot c_i[\Delta_i^n].
\end{aligned}
\tag{7}
$$

### B.2  STACKED DISE

An extension of DISE to increase its depth would consist of just stacking recurrent neural networks on top of each other (Hermans & Schrauwen, 2013). We modify this standard approach with the goal of reinforcing time information into higher layers. To do so we also apply gating mechanisms at each layer. The time-enriched representations at the $l$-th layer are defined as:

$$
\begin{aligned}
\hat{o}_i[\Delta_i^n]^{(l)} &= o_i[\Delta_i^n]^{(l)} \odot \sigma(W_\delta^{(l)} f_{\text{enc}}^{\delta^F}(\delta_{i;n}^F)) \odot \sigma(W_{\text{t}}^{(l)} f_{\text{enc}}^{\text{t}}(t_{i+n}^a)) \\
\hat{h}_{i-1}[\Delta_i^n]^{(l)} &= h_{i-1}[\Delta_{i-1}^1]^{(l)} \odot \sigma(V_\delta^{(l)} f_{\text{enc}}^{\delta^F}(\delta_{i;n}^F)) \odot \sigma(V_{\text{t}}^{(l)} f_{\text{enc}}^{\text{t}}(t_{i+n}^a))
\end{aligned}
\tag{8}
$$

where the input at the $l$-th layer is defined as

$$
o_i[\Delta_i^n]^{(l)} = \begin{cases} f_{\text{enc}}^{\circ}(o_i) & l = 0 \\ h_i[\Delta_i^n]^{(l-1)} & l > 0 \end{cases}
\tag{9}
$$

and the $W^{(l)}$, $V^{(l)}$ matrices map all latent representations to the same dimension. A prediction is computed as $f_\theta(h_i[\Delta_i^n]^{(L)})$, where $L$ is the number of layers.

### B.3  RISE2DISE

RISE2DISE defines discount factors similar to Equation (3) that are computed with respect to future observed values: $\gamma_{i;n}^o = \gamma^o(\delta_{i;n}^F)$ and $\gamma_{i;n}^h = \gamma^h(\delta_{i;n}^F)$. Since it is not able to incorporate date-formatted

timestamps, $\Delta_i^n$ simply amounts to $(\delta_{i;n}^F)$. It learns time-enriched representations for input and hidden state as follows:

$$
\begin{aligned}
\hat{o}_i[\Delta_i^n] &= g_o(o_i, \delta_{i;n}^F) = \gamma_{i;n}^o o_i + (1 - \gamma_{i;n}^o) o_{av_{i'}} \\
\hat{\boldsymbol{h}}_{i-1}[\Delta_i^n] &= g_h(\boldsymbol{h}_{i-1}[\Delta_{i-1}^1], \delta_{i;n}^F) = \boldsymbol{h}_{i-1}[\Delta_{i-1}^1] \odot \gamma_{i;n}^h,
\end{aligned}
\tag{10}
$$

where $o_{av_{i'}}$ is the average of the values prior to $t_i$ of vector $\mathbf{o}$.

## C  TRAINING DETAILS

We use the same evaluation protocol and setup as that of Fox et al. (2018). We stack two GRUs of 512 hidden units $d_h$ for all approaches. When using the multi-classification formulation, the probability distributions outputted by the function $f_\theta$ are translated to predictions by taking the value represented by the class with maximum probability. Models are evaluated at any point in time in which there are at least ten samples of prior data. We validate performance according to the median APE on the validation set after every epoch. We validate the weight decay between the values $\{10^{-2}, 10^{-3}\}$. Models were trained for 100 epochs, but usually the best validated model is obtained within the first 50 epochs. All remaining model details, such as the initialization procedure and the initial learning rate for ADAM (Kingma & Ba, 2015), used the Tensorflow (Abadi et al., 2016) default values. The same evaluation protocol and setup has been followed for all trained models.

When using the digit-level GRU $f_{\mathtt{gru}}$ as encoding function, we embed tokens of the vocabulary into a 64-dimensional space, and the number of hidden units $d_\Omega$ of the standard GRU is validated between $\{64, 128\}$. To avoid increasing complexity during training this hyperparameter is validated globally, and not individually for each dedicated encoding function. Similarly, the dimension $d_\Omega$ of the feedforward encoders is validated between $\{64, 128\}$.

## D  EXPERIMENTAL SECTION: NUMERICAL AND VISUAL INFORMATION

### D.1  DATASETS

**Blood Glucose**  This constitutes the main benchmark data set of our experiments. We run experiments in this data set for the time horizon prediction problem, as the preprocessed data set is free of missing values. We also run experiments in this data set for the next observed value prediction problem. We assume the existence of a probability distribution $P(\delta)$ that determines the time gap $\delta$, measured in sampling periods, between two observed values. This probability distribution is chosen so as to mimic the presence of blocks of consecutive missing values, which is the main driving factor in the design of all above-mentioned methods. Thus, we draw values from an exponential distribution $P(\delta) = \frac{1}{\beta} \exp\left(-\frac{\delta - 1}{\beta}\right)$ (with $\delta \geq 1$) to determine the time gap in terms of sampling periods from one observation to the next one. Thus, all intermediate observations are dropped. The hyperparameter $\beta$ controls the average time gap between pairs of observations, the larger the value is, the larger the average time gap between observations is.

Information about time gap values between consecutive observations is displayed in Figure 6. While in the blood glucose data sets these values have been perfectly generated by an exponential distribution, one can observe the time gaps for the air pollution data sets are more irregularly distributed.

Following Fox et al. (2018), where blood glucose measurements were split into chunks of 101 data points, we also split the air pollution time series into chunks of 96 measurements (4 days).

Statistics of the data sets can be found in Table 5.

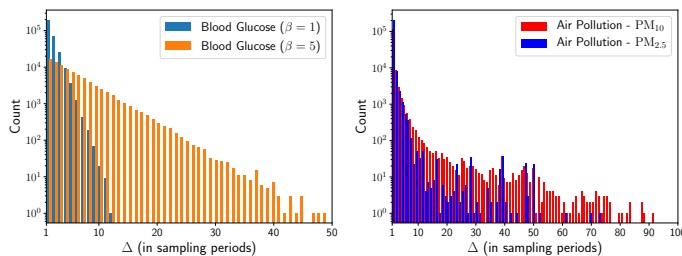

Figure 6: Histogram of time gap values between consecutive non-missing observations.

| Data set | # Observed Values [%] | # Missing Values [%] | Range | Sampling Period |
|---|---|---|---|---|
| Blood Glucose | 616,043 [100%] | 0 [0%] | 40 - 400 | 5 minutes |
| Blood Glucose ($\beta = 1$) | 372,404 [61%] | 243,639 [39%] | 40 - 400 | 5 minutes |
| Blood Glucose ($\beta = 5$) | 106,975 [18%] | 509,068 [82%] | 40 - 400 | 5 minutes |
| Air Pollution - $PM_{2.5}$ | 273,553 [87%] | 41,807 [13%] | 1 - 1,000 | 1 hour |
| Air Pollution - $PM_{10}$ | 173,243 [55%] | 142,117 [45%] | 5 - 1,000 | 1 hour |

Table 5: Data set statistics.

## D.2 EXPERIMENTS

For the Blood Glucose ($\beta = 5$) data set, we show the correlation matrices of the representations learned for a range of values of the input and time gap signals in Figure 7. For small values of the input signal the respective encoding function learns dedicated representations, whereas for larger values the representations are more correlated. Interestingly, the correlation matrix of the time gap representations can be divided into 3 block matrices: one for short-term predictions (up to 10 minutes), one for mid-term predictions (from 10 to 50 minutes) and one for long-term predictions (from 50 minutes onwards).

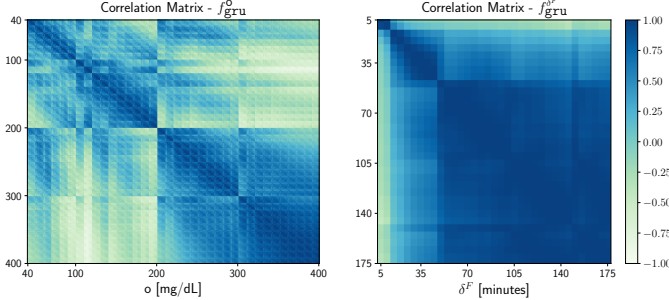

Figure 7: (Left) Correlation matrix of the representations learned by the encoding functions $f_{\mathtt{gru}}^{\circ}$ and (Right) $f_{\mathtt{gru}}^{\delta^F}$ in Blood Glucose ($\beta = 5$).

In both correlation matrices one may observe some abrupt transitions. As an example, for values of the input signal before and after 200 mg/dL the learned representations are negatively correlated. Similarly, when the time gap between two observations takes values before and after 50 minutes—this amounts to 10 once normalized by the sampling period—the correlation is relatively small. Interestingly, in both cases this occurs when there is a change in the leftmost digit of the value. Therefore, one may wonder whether the representations learned by the encoding function $f_{\mathtt{gru}}$ are driven by the atomic units the numerical data is decomposed into. We run experiments where numerical data is converted into hexadecimal and decomposed as a sequence of elements of the vocabulary $\{0 - 9, \text{a-f}\}$. Decimal and hexadecimal notation schemes are compared in Table 6 in all blood glucose problems.

The correlation matrices obtained with a hexadecimal notation are depicted in Figure 8. While visually similar to those of Figure 7, one may observe some differences: the correlation matrix for the input signal is very smooth, being seemingly unaffected by changes in the leftmost digit; while

| Notation
Data | Decimal | Hexadecimal |
|---|---|---|
| Blood Glucose (Horizon Prediction) | **3.86** \| **7.20** | 3.89 \| 7.25 |
| Blood Glucose ($\beta = 1$) | 2.06 \| 3.97 | **2.02** \| **3.88** |
| Blood Glucose ($\beta = 5$) | **5.51** \| 11.29 | 5.61 \| **10.92** |

Table 6: Blood Glucose: DISE-GRU [R]'s performance with decimal and hexadecimal notation.

the correlation matrix for the time gap signal also presents 3 block matrices, the transition from the second to the third block is much smoother than that of its counterpart with a decimal notation.

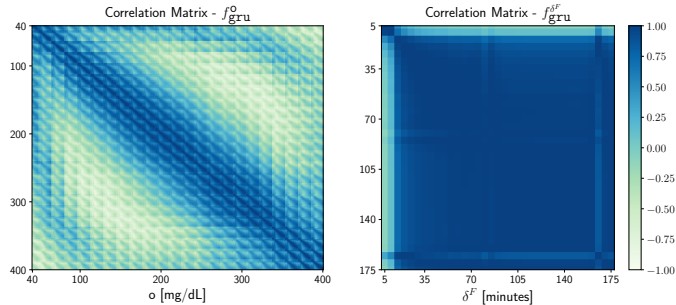

Figure 8: (Left) Correlation matrix of the representations learned by the encoding functions $f^{\circ}_{\text{gru}}$ and (Right) $f^{\delta^F}_{\text{gru}}$ in Blood Glucose ($\beta = 5$). Numerical data is converted into hexadecimal.

Experiments in the air pollution data sets have shown that timestamps have a large positive impact to model abrupt changes in the bias of the signal. The correlation matrices of the representations learned for the timestamps $t^a$ are shown in Figure 9. Most of the values in the correlation matrix for the particulate $PM_{2.5}$ are in the range between 0.95 and 1. On the other side, the correlation matrix for $PM_{2.5-\text{peak}}$ is directly interpretable and matches the prior knowledge we have of the data. The encoding function $f^{t}_{\text{gru}}$ learns dedicated representations for the 2nd and 15th day of the month, which will serve to allocate some dimensions of the input and state representation to handle this property of the signal.

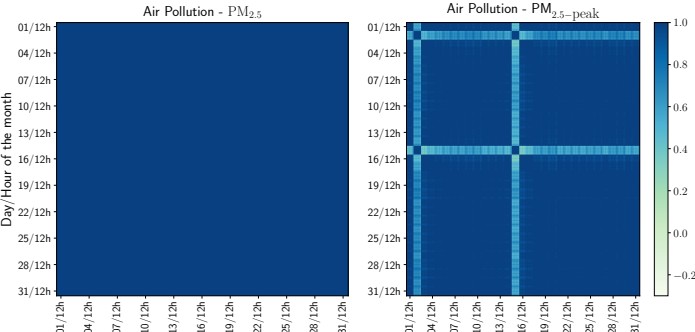

Figure 9: (Left) Correlation matrix of the representations learned by the encoding function $f^{t}_{\text{gru}}$ in $PM_{2.5}$ and (Right) $PM_{2.5-\text{peak}}$.

