# OpenReview forum: "RISE and DISE: Two Frameworks for Learning from Time Series with Missing Data"
_ICLR.cc/2020/Conference — Reject_

### Official Review · AnonReviewer1 · 2019-10-16
**Official Blind Review #1**

**Rating:** 3

**Review:**

The authors propose a new univariate time series analysis framework called RISE, which unifies the existing work on adapting RNNs to irregular time series. Building on top of the RISE, they propose a modification called DISE in which the algorithm skips the intervals without any observations. In that sense, DISE can be considered a marked point process analysis algorithm. They quantify the performance of the RISE and DISE on two datasets.

Table 1 is a valuable summary of the existing efforts on adapting RNNs to irregular time series. However, the paper overstates its scope. This work only studies RNNs. There are many alternatives for analysis of irregular time series, including Gaussian processes [1], ordinary differential equations [2], convolutional neural networks [3], neural point processes and spiking neural networks [4]. These references are only notable examples from each category and there are many more.

A major limitation of this paper is that it only applies to the univariate time series. Usually in domains such as healthcare, almost always different variables have different missing rates. Multiple works address the multivariate case, see [5] for example.

The main dataset used for evaluation is the Glucose dataset. However, this dataset is a peculiar and very specific dataset because its goal is to predict glucose level for type-1 diabetics only base on the past glucose measurements. While this task is meaningful, human biology states that forecasting glucose levels without knowledge of insulin injection or carbohydrate consumption is an extremely difficult task. In this setting, the most useful data is the latest data point. This dataset is an extreme forecasting task in absence of major predictors and I do not think it should be the primary dataset for evaluation of a new algorithm.

Finally, the main idea of skipping the intervals without measurements is not very novel given the existing literature on neural point processes. Also, it is not enough contribution for a full conference paper.

[1] Shukla, Marlin (2019) Interpolation-Prediction Networks for Irregularly Sampled Time Series. In ICLR.

[2] Chen, T. Q., Rubanova, Y., Bettencourt, J., & Duvenaud, D. K. (2018). Neural ordinary differential equations. In NeurIPS.

[3] Nguyen, P., Tran, T., Wickramasinghe, N., & Venkatesh, S. (2016). Deepr: a convolutional net for medical records. IEEE BHI.

[4] Islam, K. T., Shelton, C. R., Casse, J. I., & Wetzel, R. (2017). Marked point process for severity of illness assessment. In Machine Learning for Healthcare Conference.

[5] Che, Z., Purushotham, S., Li, G., Jiang, B., & Liu, Y. (2018). Hierarchical deep generative models for multi-rate multivariate time series. In ICML.

**Experience Assessment:**

I have published one or two papers in this area.

**Review Assessment: Checking Correctness Of Derivations And Theory:**

I assessed the sensibility of the derivations and theory.

**Review Assessment: Checking Correctness Of Experiments:**

I assessed the sensibility of the experiments.

**Review Assessment: Thoroughness In Paper Reading:**

I read the paper at least twice and used my best judgement in assessing the paper.

---

> ### Author Response · Authors · 2019-11-15
> **Rebuttal Review #1**
>
> We thank the reviewer for the comments.
>
> We agree with the reviewer that the scope of the paper concerns RNN-based models. We will include pointers and discussions to some of the works that the reviewer suggests. One of the goals of the paper is to show that an important number of recent RNN-based methods to learn from time series with missing data are instances of the framework introduced by us called RISE. This helps to understand the nuances across all these methods. Then we propose a framework wherein it only learns from observed data and the discount factors are to be learned. We do not claim that RNN-based models are the only type of solutions to address this problem.
>
> We already acknowledge in the paper that the proposed instances of DISE are not able to handle multivariate time series, and mentioned that future work should address this limitation. However, there exist many problems/data sets, such as the glucose prediction data set in [Fox et al. 2018], that consist of uni-variate time series.
>
> For the sake of completeness, not only do we compare to [Fox et al. 2018] in the time horizon prediction problem, but also evaluate in a forecasting problem where there are missing data points. We agree that the latter problem is difficult, but we still show significant improvements over the baselines. Moreover, for the same problem we benchmark all methods in two more uni-variate time series. We think that overall the experiments are appropriate in terms of quantity and variety.

---

### Official Review · AnonReviewer3 · 2019-10-23
**Official Blind Review #3**

**Rating:** 1

**Review:**

The paper studies missing value imputation in univariate time series. The paper introduces a framework called RISE which provides a unified framework for existing methods in the domain. The authors further propose DISE which generalizes RISE. Experiments on time series forecasting demonstrate improved performance.

+ It is quite interesting that the unified framework RISE can encompass the existing missing value imputation methods as special cases.
+ The idea of using learnable factors for relative and absolute time information in DISE makes sense.

- The alternative notations for the proposed framework and the existing framework are very confusing. It is not clear why such alternative notations for the same setting are necessary.
- The modeling novelty is quite limited. Other than learnable factors for absolute and relative time information, there is very little motivation or theory regarding the modeling choice.
- The experiments are not well motivated. The paper compares with a lot of missing value imputation baselines but the experimental setup is actually for extrapolation. In this case, a more proper set of baselines should be time series forecasting methods for irregularly sampled data such as Phased LSTM.


**Experience Assessment:**

I have published in this field for several years.

**Review Assessment: Checking Correctness Of Derivations And Theory:**

I carefully checked the derivations and theory.

**Review Assessment: Checking Correctness Of Experiments:**

I carefully checked the experiments.

**Review Assessment: Thoroughness In Paper Reading:**

I read the paper thoroughly.

---

> ### Author Response · Authors · 2019-11-15
> **Rebuttal Review #3**
>
> We thank the reviewer for his/her comments, although we respectfully disagree with the overall score.
>
> Coming up with a single notation to describe both frameworks is possible, but this comes at the price of overusing sub/superscripts to accurately represent the data in both frameworks. With the goal of alleviating the already heavy notation, we intentionally propose an alternative notation.
>
> We show empirically that learning discount factors is an important contribution, as they lead to improvements in several problems and data sets. As opposed to previous work where such factors have always a predefined (exponential) form, we are the first ones in proposing to learn such factors. Moreover, to our knowledge we are the first ones in making use of absolute time information, whose benefits are showcased in Table 4 and Figure 4.
>
> We split Table 1 into methods (upper part) that are suitable for extrapolation/forecasting and methods (lower part) that are suitable for imputation, all of them published recently in top-tier conferences. As the evaluation tasks are forecasting problems we compare to all methods summarized in the upper part of the table. These methods can also be used as imputation methods, but their clear disadvantage with respect to imputation methods (lower part of Table 1) is that they do not make use of future information.

---

### Official Review · AnonReviewer2 · 2019-10-24
**Official Blind Review #2**

**Rating:** 3

**Review:**

Overall:
Provides a nice summary of different methods for dealing with missing values in neural net time series models and proposes a new technique that does not involve running a series of possibly diverging predictions but rather jumps ahead to reason about arbitrary points in the future in a “single hop”, avoiding the risks associated with compounding errors. Also proposes a new method for encoding values that’s quite unusual but appears to work very well.

Overall, the paper is mostly clear, the technique is reasonable, and the best model does indeed appear to work well. I have only one serious reservation about this paper - and it is an extremely serious concern about the experimental setup, and I would ask that the authors clarify this point for me in a response. DISE works poorly or only comparably well to the baselines in all tasks unless the GRU-based input encoding is used. Obviously any RISE model likewise depends on an input encoding, so the question is whether the baseline RISE models were given the benefit of the GRU-based input encoding. If not, please provide this comparison.

Minor comments:
“replace the standard input with a transformed input ˆx” -> I find this wording awkward. If the input is missing, it cannot be transformed, it can be predicted using a conditional model over data given a representation of past (and/or future) observations, or it can be a (probably learned) dummy value, but please clarify this wording- it’s essential.

Similarity of DISE to prior work:
There are a number of processes that build representations based on measurements that happen at random times without “rolling forward” a single step model, for instance, the neural Hawkes process (Mei and Eisner, 2017 or so), which has also been applied to impute missing values. Some discussion of the relationship to work like this is recommended. Additionally, the idea of learning representations based on predicting values at several time scales into the future comes up in contrastive predictive coding (van Oord et al, 2018).

**Experience Assessment:**

I have read many papers in this area.

**Review Assessment: Checking Correctness Of Derivations And Theory:**

N/A

**Review Assessment: Checking Correctness Of Experiments:**

I carefully checked the experiments.

**Review Assessment: Thoroughness In Paper Reading:**

I read the paper at least twice and used my best judgement in assessing the paper.

---

> ### Author Response · Authors · 2019-11-15
> **Rebuttal Review #2**
>
> 1) We thank the reviewer for the comments. The reviewer made a good point regarding the usage of our encoding function into instances of RISE. Unfortunately, we did not have enough time to properly address this question for the rebuttal, but we will do it in a future resubmission
>
>
> 2) We also thank the reviewer for the pointers to the related work. We will include them in the paper.
> The neural Hawkes process closely relates to some of the works discussed in this submission. Under some modifications it resembles the baseline referred to as RISE2DISE, where i) the past history is weighted by an exponential decay mechanism, and ii) it only learns from observed data.
>
> We relate our work to existing approaches that learns several time dependencies (Pachitariu et al. 2013; Chang et al. 2017), but we agree that the work by van Oord et al. should also be included in the discussion. Different to all these works, our approach is suitable for irregularly spaced time series.
>
> 3) Regarding the sentence “replace the standard input with a transformed input”, the reviewer is completely right. That is what these instances do. Table 1 provides a summary of the transformed input \hat{x} for a number of RISE instances. However, it is true that the used term may not be 100% accurate, as the reviewer suggests.
>
> We really appreciate all the points raised by the reviewer.

---

### Decision · Program_Chairs · 2019-12-19

**Decision:**

Reject

**Comment:**

The paper attacks the important problem of learning time series models with missing data and proposes two learning frameworks, RISE and DISE, for this problem. The reviewers had several concerns about the paper and experimental setup and agree that this paper is not yet ready for publication. Please pay careful attention to the reviewer comments and particularly address the comments related to experimental design, clarity, and references to prior work while editing the paper.